# Non-Invasive Biomarkers in the Diagnosis of Upper Urinary Tract Urothelial Carcinoma—A Systematic Review

**DOI:** 10.3390/cancers14061520

**Published:** 2022-03-16

**Authors:** Łukasz Białek, Konrad Bilski, Jakub Dobruch, Wojciech Krajewski, Tomasz Szydełko, Piotr Kryst, Sławomir Poletajew

**Affiliations:** 1Department of Urology, Centre of Postgraduate Medical Education, 01-813 Warsaw, Poland; konradbilski@gmail.com (K.B.); jdobruch@cmkp.edu.pl (J.D.); 2Department of Minimally Invasive and Robotic Urology, Wroclaw Medical University, 50-556 Wrocław, Poland; wk@softstar.pl (W.K.); tomasz.szydelko@umw.edu.pl (T.S.); 3Second Department of Urology, Centre of Postgraduate Medical Education, 01-813 Warsaw, Poland; piotr.kryst@cmkp.edu.pl (P.K.); slawomir.poletajew@cmkp.edu.pl (S.P.)

**Keywords:** upper urinary tract urothelial carcinoma, biomarkers

## Abstract

**Simple Summary:**

Laboratory and imaging tests as well as endoscopic procedures are standard diagnostic tools for the diagnosis of upper urinary tract urothelial carcinoma. We aimed to determine the effectiveness of non-invasive urinary and blodd-based tests in the diagnosis of urothelial carcinoma of upper urinary tract. Among 10,084 screened publications, 25 were eligible and included in the analysis. Most of them were conducted on small samples of patients and the control groups were heterogenous. The test used in the largest number of studies was voided urinary cytology, which has poor sensitivity with favorable specificity. Fluorescence in situ hybridization in diagnostic cytology showed higher sensitivity with equally good specificity. There were also studies on the use of tests known to diagnose bladder cancer such as NMP22, uCYT or BTA test. Other urine or blood tests have been the subject of only isolated studies, with varying results. To conclude, currently there is a lack of high-quality data that could confirm good effectiveness of non-invasive tests used in the diagnosis of upper urinary tract urothelial carcinoma.

**Abstract:**

Beyond laboratory, imaging and endoscopic procedures, new diagnostic tools are increasingly being sought for the diagnosis of upper urinary tract urothelial carcinoma (UTUC), especially those that are non-invasive. In this systematic review, we aimed to determine the effectiveness of non-invasive tests in the diagnosis of UTUC. PubMed and Embase electronic databases were searched to identify studies assessing effectiveness of non-invasive tests in the primary diagnosis of UTUC. The study protocol was registered with PROSPERO (CRD42020216480). Among 10,084 screened publications, 25 were eligible and included in the analysis. Most of them were conducted on small samples of patients and the control groups were heterogenous. The test used in the largest number of studies was voided urinary cytology, which has poor sensitivity (11–71.1%) with favorable specificity (54–100%). Fluorescence in situ hybridization in diagnostic cytology showed higher sensitivity (35–85.7%) with equally good specificity (80–100%). There were also studies on the use of tests known to diagnose bladder cancer such as NMP22, uCYT or BTA test. Other urine or blood tests have been the subject of only isolated studies, with varying results. To conclude, currently there is a lack of high-quality data that could confirm good effectiveness of non-invasive tests used in the diagnosis of UTUC.

## 1. Background

Upper urinary tract urothelial carcinoma (UTUC) is an uncommon neoplasm and accounts for only 5–10% of all urothelial cancers. It is defined as urothelial malignancy originating from epithelial cells lining the renal collecting system or ureter. Despite similarities in histology and etiology, differences in the diagnostic and treatment process between UTUC and urothelial bladder cancer (UBC) result in these tumors sometimes being referred to as ‘disparate twins’ [1]. These similarities are on the one hand providing advantages, as some diagnostic and treatment methods can be adapted more easily from UBC, but on the other hand, disregarding the characteristic differences between UTUC and UBC may result in treatment failure. UBC and UTUC share some common risk factors but also exhibit significant differences in the prevalence of common genomic alterations [2]. However, it is more likely for UBC to develop after UTUC (22–47%) than UTUC to develop after UBC (2–6%) [3]. Moreover, many cases of UTUC are genetically linked to hereditary non-polyposis colorectal cancer (HNPCC), also known as Lynch syndrome [4]. The estimated annual incidence is 1–2 cases per 100,000 [5]. However, the number of new cases seems to be rising recently due to improved diagnostic imaging and endoscopic techniques [6]. Most of the patients with localized disease present with local symptoms, however one in three patients is diagnosed by accident [7].

UTUC diagnosis bases on the combination of laboratory, imaging and endoscopic procedures. This leads to a prolonged diagnostic process, exposure of the patient to radiation and often the need for surgical management [5]. Beyond standard procedures, new diagnostic tools are increasingly being sought, especially those that are non-invasive. These include fluorescent in situ hybridization (FISH) cytology, serum microRNA or urine-based tests, especially those that have been previously studied for use in the diagnosis of bladder cancer. However, their efficacy has not been accurately established.

The aim of this systematic review was to determine the potential effectiveness of non-invasive tests in the diagnosis of UTUC.

## 2. Material and Methods

The systematic review was performed in accordance with the Preferred Reporting Items for Systematic Reviews and Meta-Analyses (PRISMA) statement and the Cochrane Handbook for Systematic Reviews of Interventions. Study protocol was established in priori and was registered with PROSPERO (CRD42020216480).

A systematic search was conducted independently by two authors (Ł.B. and K.B.) through PubMed and Embase electronic databases according to the PRISMA statement [8]. The last search was performed on 1 November 2020. The search query was (biomarker OR tumor marker OR tumour marker OR assay OR test OR non-invasive) AND (upper tract urothelial carcinoma OR upper tract urothelial cancer OR upper tract tumor OR upper urinary tract OR upper tract transitional cell carcinoma OR renal pelvis tumor OR ureter tumor OR kidney pelvis tumor and UTUC). The search included articles without time limitations. Only publications in English were considered and evidence was limited to human data.

In this systematic review, studies which met the following criteria were included: studies including at least five patients with histologically confirmed primary UTUC and patients without urothelial cancer (control group); studies assessing the diagnostic performance of a test with sensitivity/specificity and/or positive predictive value (PPV)/negative predictive value (NPV) and/or area under the curve (AUC); studies assessing non-invasive tests, i.e., blood-based or voided urine-based; studies with full-text publications in English. Moreover, all references within retrieved articles were screened for additional relevant articles. Studies assessing tests based on urine from ureter or bladder washings, as not non-invasive, were excluded from analysis. Reviews, case reports, letters, and conference abstracts with no full-text or commentaries were excluded.

After removal of duplicates, two authors (ŁB and KB) independently evaluated the titles and abstracts of the retrieved records. All potentially eligible studies were evaluated as full text if available. Disagreements were resolved by consultation with the senior author (SP).

## 3. Evidence Synthesis

After screening 10,084 publications, 25 were eligible to be included in this systematic review. Figure 1 shows the selection process of the study in accordance with PRISMA. Of the 25 articles selected, the authors addressed the topic of diagnostic accuracy of voided urinary cytology (VUC) in 17 of them and fluorescence in situ hybridization in diagnostic cytology (FISH) in 14 of them. The remaining tests were the subject of one or two studies. Most of the studies were from China (8); 3 were from Japan, 3 from USA; the remaining 8 were from European countries (Germany (2); Austria (2); Serbia (2); Spain (1); Portugal (1)). The methodology of the studies is heterogenous; most of the studies comprised a small number of patients with UTUC ranging from 9 to 147. Oncological characteristics of the study group were not available in all studies.

## 4. Voided Urinary Cytology

Of the 25 articles, 17 deal with the use of VUC in UTUC diagnosis [9,10,11,12,13,14,15,16,17,18,19,20,21,22,23,24,25]. Most of the studies were conducted on small cohorts of patients. The largest number of patients was presented in study by Gomelia et al. [15], but there are no data on their oncological characteristics or information on the control group. In three studies [16,20,21], the control group consisted of patients who were originally suspected of having UTUC but were cancer-free. In these papers, sensitivity ranged from 31.6% to 60% and specificity from 80% to 100%. In other studies, the control group consisted of patients undergoing a diagnostic management of hematuria, patients with urolithiasis, BPH or other urological problems, or even healthy volunteers. Despite the high heterogeneity of the control groups, the results show a common trend. VUC for the diagnosis of UTUC has poor sensitivity (11–71.1%) with very good specificity (54–100%). Detailed information about VUC performance is available in Table 1.

## 5. Fluorescence In Situ Hybridization in Diagnostic Cytology

Fourteen studies aimed to assess the diagnostic performance of FISH [10,14,15,16,18,19,20,21,22,23,24,25,26,27] (Table 2). None of the above studies included a large group of patients. The same three papers as previously described [16,20,21] were the only ones to use as a control group patients suspected of having UTUC but in whom the cancer was eventually excluded. The results of these studies show a much higher sensitivity of FISH (73.7–87.5%) with a maintained high specificity (80–89.8%). The study by Huang et al. [27] is also noteworthy, as they evaluated the role of FISH in the diagnosis of UTUC in patients with asymptomatic hematuria and negative VUC. Although only 9 of 285 such patients had UTUC, the sensitivity of FISH for detecting UTUC was 100% and specificity was 99.3%. In other studies, the control group was heterogeneous, with FISH sensitivity ranging from 35–85.7% and specificity from 80–100%.

## 6. Other Urinary Tests

Various authors attempted to evaluate the feasibility of different tests available for usage in the diagnosis of bladder cancer. These include Bladder Tumour Antigen (BTA) test, Nuclear Matrix Protein 22 (NMP22) or Immunocyt/uCyt+.

The BTA test detects human complement factor H-related protein. It has been proved that it is secreted in urine by bladder cancer cells. Walsh et al. [9] conducted a study that evaluated the diagnostic efficacy of the BTA test in the diagnosis of urothelial carcinoma including UTUC. The majority of 27 patients with UTUC in this study were patients with non-invasive cancer of intermediate grade (G2 according to WHO 1973). The sensitivity of the test was determined to be 82% and its specificity was 89%, which was significantly better than VUC in the same group of patients (11% and 54%, respectively).

The NMP22 test is a non-invasive method for the detection of protein level of nucleus mitotic apparatus in urine sample. Healthy people usually have a very small amount of NMP in their urine, while the cells of urothelial carcinoma contain up to 80 times higher concentration of NMP than normal cells, and it is released in the urine after cell death. The efficacy of NMP 22 in the diagnosis of UTUC has been the subject of two studies [13,18]. The authors determined the sensitivity of this test to be 70–70.5% and the specificity to be 43.2–92%.

ImmunoCyt is immunocytochemical test that can detect monoclonal antibodies directed against three specific antigens, two mucins and a carcinoembryonic antigen, expressed by urothelial carcinoma. The effectiveness of this test has only been assessed in two studies that included a small group of patients. Sensitivity was estimated to be 55.6–75% and specificity 79.2–95% [11,18].

Our systematic review also uncovered single studies that evaluated the diagnostic efficacy of novel biomarkers from urine, as well as their combination with cytology. These tests typically use advanced molecular biology studies. 

Monteiro-Reis et al. in their study [28] evaluated the efficacy of DNA methylation-based biomarkers previously tested in predicting the diagnosis of bladder cancer [29]. Their proposed VIM/GDF15/TMEFF2 DNA methylation panel diagnosed UTUC with a sensitivity of 91% and specificity of 100%. p16/Ki-67 dual immunolabeling proposed by Sun et al. [10] has not produced breakthrough results in terms of effectiveness with sensitivity of 53.1% and specificity of 100%.

Another study was conducted by Yamamichi et al. [17]. The authors evaluated the diagnostic efficacy of 5-aminolevulinic acid induced (5-ALA) fluorescent urine cytology for detecting urothelial carcinoma including UTUC. Their study included 76 patients with UTUC and 158 patients with other urological conditions as controls. The sensitivity for detecting UTUC was 90.8% and specificity was 96.2%. In the same group of patients, results of conventional VUC were as follows: sensitivity 71.1%, specificity 95.6%. Moreover, it is worth noting the excellent sensitivity of 5-ALA in almost all subgroups, including patients with low-grade tumors, at 91.5%.

Hayashi et al. [12] conducted a study in which they evaluated the diagnostic efficacy of performing a rtPCR test to look for hotspot mutations of telomerase reverse transcriptase (TERT) promoter and fibroblast growth factor receptor 3 (FGFR3) in cell-free DNA in urine. The single mutation assay results, although with 100% specificity, had low sensitivity (16.1–46.4%). However, by the combination of TERT promoter and FGFR3 mutation assessment with VUC results, the authors achieved a sensitivity of 78.6% and specificity of 96%. 

Combination of VUC and BTA test was assessed by Jovanovic et al. [30]. The control group in this study comprised patients with urolithiasis. The combination of both voided urine BTA test with VUC gave the sensitivity of 81.3% and specificity of 73.3%.

Results of the performance of tests other than VUC and FISH urinary tests are summarized in the Table 3.

## 7. Blood-Based Tests

We also identified four studies assessing diagnostic performance of blood-based biomarkers (Table 4). 

MicroRNAs (miRNAs) are small, non-coding RNA molecules, which modify the expression of many human genes. They are detectable in a variety of body fluids including serum. Expression profiling studies demonstrate tumor-specific microRNA expression. Two of the studies addressed serum miRNAs as potential diagnostic biomarkers of UTUC [31,32].

In both, the groups were heterogeneous in terms of cancer stage and histological grade and also in terms of control group selection. The diagnostic efficacy results of various miRNA molecules presented in these studies are within wide ranges: sensitivity 29.5–97.8%; specificity 29.4–100%.

**Table 4 cancers-14-01520-t004:** The performance of blood-based tests in the diagnosis of UTUC.

TEST	Source	Author	UTUC pts Number	Country	M:F	Tumor Stage	Tumor Grade	Control pts Number	M:F	Control Group Characterization	Sensitivity	Specificity	PPV	NPV	AUC
diagnostic N-glycan score	serum	Tanaka [33]	55	Japan	n/a	n/a	n/a	435	218:217	122 healthy; 96 PCa	77.10	97.20	79.70	96.80	
miRNA	serum	Kriebel [31]	44	Germany	23:11	Ta–18; T1–7; T2–3; T3–15; T4–1	G1–6; G2–28; G3–10	34	23:11	BPH, urethral stricture, incontinence, stones	29.50–84.10	29.40–91.20	n/a	n/a	0.541–0.726
miRNA	serum	Tao [32]	58	China	41:17	T1–41; T2–14; T3–20; T4–3	LG 18; HG–40	42	30:12	hematuria	58.70–97.80	70–100	n/a	n/a	0.642–0.998
phosphoprotein-1	plasma	Li [34]	105	China	45:60	Ta–7; T1–41; T2–23; T3–34	LG–51; HG–54	71	n/a	10 healthy; 2 enteritidis; 6 BPH; 27 stones; 15 hydronephrosis; 6 renal cyst; 5 adrenal adenoma	n/a	n/a	n/a	n/a	0.838

Tanaka et al. presented the results of a study in which they evaluated whether the aberrant N-glycosylated serum immunoglobulins established as a diagnostic N-glycan score (dNGScore) could be applied as a diagnostic marker of urothelial carcinoma including UTUC [33]. Out of all patients with urothelial carcinoma, 55 were diagnosed with UTUC. The control group consisted of healthy volunteers and patients with prostate cancer. The dNGScore achieved good sensitivity of 77.1% with excellent specificity of 97.2%.

Another study investigating the possible use of substances contained in blood was conducted by Li et al. [34]. They found that UTUC diagnosis based on determination of plasma phospoprotein-1 levels can achieve AUC = 0.838.

## 8. Discussion

Imaging and surgical procedures such us ureterorenoscopy with biopsy and ureter washings remain the gold standard in the diagnostic process of UTUC, according to the European Association of Urology clinical guidelines [35]. Computed tomography (CT) urography shows the highest diagnostic accuracy in the evaluation of UTUC. According to the meta-analysis performed by Janisch et al., it has the sensitivity of 92% and specificity of 95% [36]. However, it is worth mentioning that even 48% of patients suspected for UTUC with ‘negative’ CT have positive findings in ureteroscopy [37]. The diagnostic performance of magnetic resonance (MR) urography is inferior to CT with the sensitivity of 75% for tumors <2 cm [38] and thus should only be used in patients who cannot undergo CT urography. Positron emission tomography is not recommended in the primary diagnosis of UTUC [35]. Moreover, it is worth mentioning that before starting any type of treatment, it should be determined whether the patient has distant metastases, as they may be present at diagnosis in about 9% of patients [39]. The imaging of choice for distant metastasis is CT, nevertheless ^18^F-Fluorodeoxglucose positron emission tomography/computed tomography has a promising sensitivity and specificity for the detection of nodal metastasis [40].

Endoscopic evaluation remains an integral part of decision-making processes in such cases. Abnormal urinary cytology results with normal cystoscopy suggests UTUC. Ureter washings cytology shows higher sensitivity and specificity than VUC [41], however such procedures include an invasive way of collecting urine.

In our systematic review we focused on non-invasive tests and biomarkers of UTUC. The results presented above emphatically demonstrate the lack of high-quality scientific evidence on this topic. In the vast majority of the studies cited, the study groups were small and the control groups were heterogeneous, often artificially formed, as these were usually patients with no suspicion of UTUC. Furthermore, the overwhelming number of articles cited deal with the use of VUC and FISH, rather than novel biomarkers. VUC, similarly to patients with bladder cancer, shows very good specificity but definitely insufficient efficacy. The sensitivity of FISH is much higher and specificity is still high. However, it is worth noting that this type of testing is characterized by high inter-observer variability [42], which results, on the one hand, from the subjective character of the evaluation and, on the other hand, from the necessary adequate experience and a large learning curve. Moreover, research is being conducted on the use of artificial intelligence and deep learning in cytopathological diagnosis. In their paper, Nojima et al. presented excellent results of this method of evaluating urinary cytology for malignant cells (with AUC 0.989) [43].

Bladder cancer and UTUC, although similar, are not identical in biological nature and prognosis (1, E). Thus, transferring the diagnostic methods of bladder cancer to UTUC is subject to risk of bias. Results of studies demonstrating the diagnostic efficacy of tests already used in the diagnosis of bladder cancer show moderate diagnostic accuracy [9,11,13,18].

Some hope may be provided by the results over the use of the concentration of certain miRNA molecules [31,32]. It seems that the rapid development of molecular biology techniques may promote the development of research in this field.

In the last years, many studies on protein markers in different types of cancer have been published using sophisticated approaches, but very little progress has been made to translate these early discoveries into clinically useful applications improving diagnosis, therapeutic choices, and monitoring. When dealing with diagnostic biomarkers, it should be remembered that there are limitations to their use. First, their diagnostic accuracy as assessed in clinical trials may be highly dependent on the selection of patients into the study and control groups, and therefore cannot always be easily translated to any patient. Second, standardization of assays and the entire pre-laboratory process is extremely important, as testing often involves the use of new biotechnology techniques. Additionally, it is important to note that when testing a biomarker, we are typically assessing the presence (or absence) of one particular trait, and the presence of cancer is associated with a number of different types of transformations, hence it seems advisable to use a multi-marker approach instead of one-marker approach whenever possible [44,45]. This should be a direction for future studies, e.g., multi-institutional studies assessing many biomarkers or panels of biomarkers, performed on a large number of patients with a reasonable and homogenous control group. Such studies are essential to introduce non-invasive diagnostics in the UTUC diagnostic process.

## 9. Conclusions

Our systematic review shows that, at this time, there is a lack of reliable and well-studied non-invasive tests that could serve in the diagnosis of UTUC. Research on this topic should continue, and, in the meantime, imaging studies and endoscopy should have a primary diagnostic role.

## Figures and Tables

**Figure 1 cancers-14-01520-f001:**
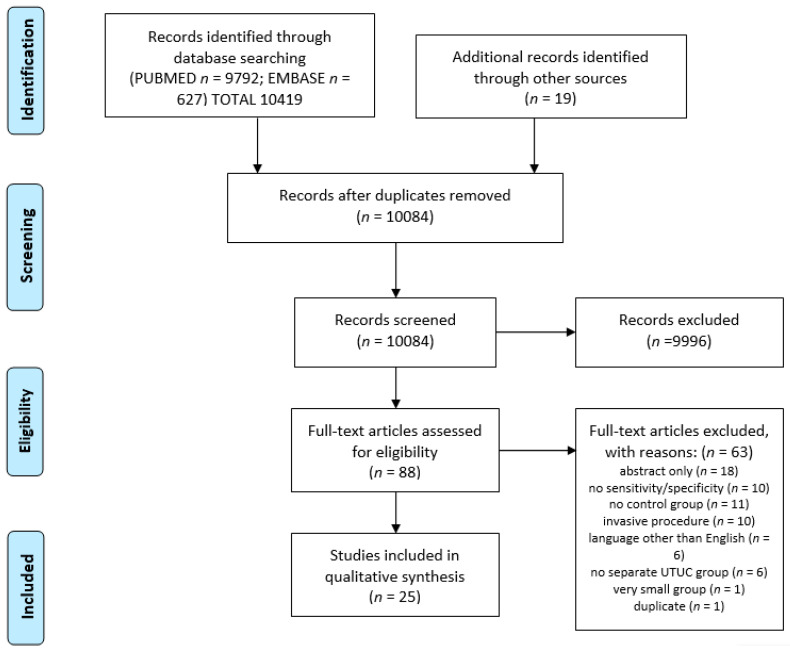
PRISMA flow diagram.

**Table 1 cancers-14-01520-t001:** The performance of VUC in the diagnosis of UTUC.

Author	UTUC pts Number	Country	M:F	Tumor Stage	Tumor Grade	Control pts Number	M:F	Control Group Characterization	Sensitivity	Specificity	PPV	NPV
Walsh [9]	27	US	20:7	Ta-16-(59.3); T1–7 (25.9); T2–3 (11.1); T3–1 (3.7)	G1–2 (7.4); G2–19 (70.4); G3–6 (22.2)	54	41:13	26 stones; 28 hematuria	11	54	11	55
Sun [10]	61	China	32:29	n/a	LG-15; HG-46	18	n/a	healthy	59	94.4	n/a	n/a
Lodde [11]	16	Austria	n/a	Ta-6; T1–2; T2 > −6	G1–3; G2–6; G3–7	21	n/a	misc.	50	100	n/a	n/a
Hayashi [12]	56	Japan	46:10	Ta or Tis-17; T1–11; T2 ± 28	LG-7; HG-47; unknown-2	n/a	n/a	hematuria, benign urological disorders, negative UC surveillance	44.6	96	92.6	60.8
Jovanovic [13]	34	Serbia	22:12	Ta-8; T1–8; T2–4; T3–13; T4–1	G1–7; G2–15; G3–12	25	08:14	stones	58.8	96	95.2	63.1
Shan [14]	50	China	32:18	Ta-6; Tis–3; T1–9; T2–9; T3–22; T4–1	G1–10; G2–13; G3–27	25	16:9	healthy	40	96	n/a	n/a
Gomella [15]	147	US	n/a	n/a	n/a	92	n/a	n/a	42.2	90.2	87.3	49.4
Akkad [16]	9	Austria	5:4	Ta-2; Tis–1; T1–1; T2–1; T3–4	G2–5; G3–4	7	3:4	suspected for UTUC but T0	60	80	75	66
Yamamichi [17]	76	Japan	n/a	n/a	n/a	158	n/a	117 BPH; 18 stones; 17 infection; 6 other	71.1	95.6	88.5	87.3
Todenhoefer [18]	62	Germany	n/a	n/a	n/a	n/a	n/a	hematuria	67.7	82.8	n/a	n/a
Chen [19]	23	US	n/a	n/a	LG-15; HG-8	5	n/a	n/a	13	80	n/a	n/a
Yu [20]	19	China	15:4	Ta-6; Tis–3; T1–5; T2–5;	G1–7; G2–8; G3–4	98	n/a	suspected for UTUC but T0	42.1	94.9	100	90.3
Lin [21]	19	China	n/a	n/a	n/a	46	n/a	suspected for UTUC but T0	31.6	100	n/a	n/a
Xu [22]	71	China	54:17	Ta-34; T1–12; T2–17; T3–6; T4–2	LG-39; HG-32	45	n/a	15 UTI; 20 stones; 10 healthy	45.1	100	n/a	n/a
Yu [23]	44	China	25:19	Ta-1; T1-6; T2–24; T3–7; T4–5	LG-14; HG-30	26	n/a	stones or BPH	27.3	100	n/a	n/a
Marin-Aguilera [24]	30	Spain	25:5	Ta-5; Tis–1; T1–5; T2–5; T3–10; Tx-4	LG-5; HG-25	19	n/a	healthy	36	100	n/a	n/a
Luo [25]	21	China	12:9	Ta-3; T1–2; T2–7; T3–7; T4–2	LG-8; HG-13	10	n/a	healthy	23.8	100	n/a	n/a

**Table 2 cancers-14-01520-t002:** The performance of FISH in the diagnosis of UTUC.

Author	UTUC pts Number	Country	M:F	Tumor Stage	Tumor Grade	Control pts Number	M:F	Control Group Characterization	Sensitivity	Specificity	PPV	NPV
Sun [10]	61	China	32:29	n/a	LG-15; HG-46	10	n/a	healthy	67.2	90	n/a	n/a
Wang [26]	34	China	26:8	Ta-1; T1–15; T2–6; T3-8; T4-4	LG-14; HG-20	33	21:12	hematuria	73.5	93.9	92.6	77.5
Shan [14]	50	China	32:18	Ta–6; Tis–3; T1–9; T2–9; T3–22; T4–1	G1-10; G2-13; G3-27	25	16:9	healthy	84	96	n/a	n/a
Gomella [15]	52	US	n/a	n/a	n/a	28	n/a	n/a	51.9	89.3	90	50
Huang [27]	9	China	8:1	T1–4; T2–4; T3–1	G1-1; G2-2; G3-6	276	140:136	asymptomatic hematuria and negative VUC	100	99.3	81.8	100
Akkad [16]	9	Austria	5:4	Ta–2; Tis–1; T1–1; T2–1; T3–4	G2-5; G3-4	7	3:4	suspected for UTUC but T0	87.5	80	87.5	80
Todenhoefer [18]	26	Germany	n/a	n/a	n/a	n/a	n/a	hematuria	61.5	80.1	n/a	n/a
Chen [19]	20	US	n/a	n/a	LG-14; HG-6	5	n/a	n/a	35	80	n/a	n/a
Yu [20]	19	China	15:4	Ta–6; Tis–3; T1–5; T2–5;	G1-7; G2-8; G3-4	98	n/a	suspected for UTUC but T0	84.2	89.8	80	97.8
Lin [21]	19	China	n/a	n/a	n/a	46	n/a	suspected for UTUC but T0	73.7	89.1	n/a	n/a
Xu [22]	71	China	54:17	Ta–34; T1–12; T2–17; T3–6; T4–2	LG-39; HG-32	45	n/a	15 UTI; 20 stones; 10 healthy	78.9	97.8	n/a	n/a
Yu [23]	44	China	25:19	Ta–1; T1–6; T2–24; T3–7; T4–5	LG-14; HG-30	26	n/a	stones or BPH	79.5	96.2	n/a	n/a
Marin-Aguilera [24]	30	Spain	25:5	Ta–5; Tis–1; T1–5; T2–5; T3–10; Tx–4	LG-5; HG-25	19	n/a	healthy	76.7	94.7	n/a	n/a
Luo [25]	21	China	12:9	Ta–3; T1–2; T2–7; T3–7; T4–2	LG-8; HG-13	10	n/a	healthy	85.7	100	n/a	n/a

**Table 3 cancers-14-01520-t003:** The performance of other urinary tests in the diagnosis of UTUC.

TEST	Author	UTUC pts Number	Country	M:F	Tumor Stage	Tumor Grade	Control pts Number	M:F	Control Group Characterization	Sensitivity	Specificity	PPV	NPV
BTA stat	Walsh [9]	27	US	20:07	Ta–16- (59.3); T1–7 (25.9); T2–3 (11.1); T3–1 (3.7)	G1–2 (7.4); G2–19 (70.4); G3-6 (22.2)	54	41:13:00	26 stones; 28 hematuria	82	89	79	91
NMP22	Jovanovic [13]	34	Serbia	22:12	Ta–8; T1–8; T2–4; T3–13; T4–1	G1–7; G2–15; G3–12	25	08:14	stones	70.5	92	92.3	69.6
NMP22	Todenhoefer [18]	20	Germany	n/a	n/a	n/a	n/a	n/a	hematuria	70	43.2	n/a	n/a
uCyt	Lodde [11]	16	Austria	n/a	Ta–6; T1–2; T2 > −6	G1–3; G2–6; G3–7	21	n/a	misc.	75	95	n/a	n/a
uCyt	Todenhoefer [18]	9	Germany	n/a	n/a	n/a	n/a	n/a	hematuria	55.6	79.2	n/a	n/a
VIM/GDF15/TMEFF2 methylation panel	Monteiro-Reis [28]	22	Portugal	12:10	Ta–15; T1 ± 7	LG–3; HG–19	20	11:09	3 healthy; 10 RCC; 7 PCa	91	100	100	91
p16/Ki-67 dual immunolabelling	Sun [10]	32	China	16:16	n/a	LG–8; HG–24	9	n/a	healthy	53.1	100	n/a	n/a
FGFR3 mutation	Hayashi [12]	56	Japan	46:10:00	Ta or Tis–17; T1–11; T2 ± 28	LG–7; HG–47; unknown-2	n/a	n/a	hematuria, benign urological disorders, negative UC surveillance	16.1	100	100	51.5
TERT promoter + FGFR3	Hayashi [12]	56	Japan	46:10:00	Ta or Tis–17; T1–11; T2 ± 28	LG–7; HG–47; unknown-2	n/a	n/a	hematuria, benign urological disorders, negative UC surveillance	55.4	100	100	66.7
TERT promoter mutation	Hayashi [12]	56	Japan	46:10:00	Ta or Tis–17; T1–11; T2 ± 28	LG–7; HG–47; unknown-2	n/a	n/a	hematuria, benign urological disorders, negative UC surveillance	46.4	100	100	62.5
VUC + TERT + FGFR3	Hayashi [12]	56	Japan	46:11:00	Ta or Tis–17; T1–11; T2 ± 29	LG–7; HG–47; unknown-2	n/a	n/a	hematuria, benign urological disorders, negative UC surveillance	78.6	96	95.7	80
BTA bard + VUC	Jovanovic [30]	35	Serbia	22:13	n/a	n/a	35	n/a	stones	81.3	73.3	76.5	83.3
5-ALA VUC	Yamamichi [17]	76	Japan	n/a	n/a	n/a	158	n/a	117-BPH; 18-stones; 17-infection; 6-other	90.8	96.2	92	95.6

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
