# Peer review of "Non-Invasive Biomarkers in the Diagnosis of Upper Urinary Tract Urothelial Carcinoma—A Systematic Review"

_cancers, 2022, doi:10.3390/cancers14061520_

Round 1
Reviewer 1 Report
A good comprehensive attempt to write up (systematic review) on upper tract urothelial cancer (UTUC). A few suggestions which should be easily addressable by the authors:
In Introduction: Authors should define what exactly is UTUC in order to differentiate it from Bladder cancer (major urothelial cancer lesion) very clearly. It is not very clear and also why it is important to differentiate between both of them in terms of treatment and prognosis and diagnostic challenges.
In Materials and Methods: An illustration/ like graphical abstract depicting different regions and sampling from different regions in terms of biomarkers described in the manuscript is highly recommended.
Discussion: Limitation of liquid or non-invasive biomarkers have been touched but needs to be described a bit more. Also, future direction of the biomarkers and the field where it is heading needs to be mentioned.
Author Response
Thank you very much for your kind review.
According to your suggestions, we have implemented the information about the definition of UTUC and the differences between bladder cancer and UTUC into the Introduction section.
The pointer to the origin of the articles included in the systematic review appears to be indeed very pertinent - thank you! We have therefore summarized the origin of the articles - this is included in a shortened version in the text in Material and Methods, and we have additionally added a column in the tables where the country of origin of the study is listed.
In the Discussion section, we elaborated on the limitations associated with biomarkers and future research directions for non-invasive diagnostic tests to detect UTUC.
Reviewer 2 Report
The authors conducted a systematic review to investigate non-invasive biomarkers in the diagnosis of upper urinary tract urothelial carcinoma.
This is a nicely written paper and the results and discussion are appropriate. One minor comment is that, although non-diagnostic, the authors could spend a couple of lines discussing in-brief the role of imaging studies (CT/MRI/PET) in UTUC detection.
Author Response
Thank you very much for your kind review.
According to your suggestion, we have implemented into the discussion some information about imaging studies in the diagnostics of UTUC.
Reviewer 3 Report
A description of the different techniques should be helpful to understand the feasible and applicability of these diagnostic options?
The authors should explain the limitation and potential complication of the techniques
Are there any of these new techniques compared with “gold standard” (ureterorenoscopy with biopsy and ureter washings) or between them??
Author Response
Thank you very much for your kind review.
According to your suggestion, we outlined briefly the essence of the tests presented in the following studies.
We also elaborated on the limitations and hassles of using biomarkers in clinical trials and in daily practice.
Answering your question about the comparison to the Gold standard - URS + biopsy - In all of the studies included in the systematic review, histopathologic confirmation, whether by URS + biopsy or by specimen after radical nephroureterectomy, partial ureterectomy, or other surgical intervention, was required for a patient to be considered to have UTUC - it was one of the inclusion criteria. However, the focus of these studies was not on comparing biomarkers with URS + biopsy; rather, the focus was on evaluating the efficacy of the biomarkers themselves or possibly comparing them to each other.
Reviewer 4 Report
The Authors performed a systematic review on non-invasive biomarkers in the diagnosis of UTUC. The paper is well written and Methods are accurate. The incidence of UTUC is not high so that many studies are performed on small cohort of patients and methods differ among studies. Unfortunately the prospective for a non-invasive diagnosis of UTUC is still far away and the priority for pubblication is low
Author Response
Thank you very much for your kind review!